# Beyond Strains: Molecular Diversity in Alpha-Synuclein at the Center of Disease Heterogeneity

**DOI:** 10.3390/ijms241713199

**Published:** 2023-08-25

**Authors:** Marcelina J. Wojewska, Maria Otero-Jimenez, Jose Guijarro-Nuez, Javier Alegre-Abarrategui

**Affiliations:** Department of Brain Sciences, Imperial College London, Hammersmith Hospital, London W12 0NN, UK

**Keywords:** alpha-synuclein, strains, heterogeneity

## Abstract

Alpha-synucleinopathies (α-synucleinopathies) such as Parkinson’s disease (PD), Parkinson’s disease dementia (PDD), dementia with Lewy bodies (DLB), and multiple system atrophy (MSA) are all characterized by aggregates of alpha-synuclein (α-syn), but display heterogeneous clinical and pathological phenotypes. The mechanism underlying this heterogeneity is thought to be due to diversity in the α-syn strains present across the diseases. α-syn obtained from the post-mortem brain of patients who lived with these conditions is heterogenous, and displays a different protease sensitivity, ultrastructure, cytotoxicity, and seeding potential. The primary aim of this review is to summarize previous studies investigating these concepts, which not only reflect the idea of different syn strains being present, but demonstrate that each property explains a small part of a much larger puzzle. Strains of α-syn appear at the center of the correlation between α-syn properties and the disease phenotype, likely influenced by external factors. There are considerable similarities in the properties of disease-specific α-syn strains, but MSA seems to consistently display more aggressive traits. Elucidating the molecular underpinnings of heterogeneity amongst α-synucleinopathies holds promise for future clinical translation, allowing for the development of personalized medicine approaches tackling the root cause of each α-synucleinopathy.

## 1. Molecular Diversity 

### 1.1. Alpha-Synuclein and Alpha-Synucleinopathies

Alpha-synucleinopathies (α-synucleinopathies) are neurodegenerative diseases characterized by aggregates of the protein alpha-synuclein (α-syn) which disseminates throughout the central and peripheral nervous system. While sharing the presence of ubiquitous α-syn aggregates, alpha-synucleinopathies display a wide spectrum of clinical and pathological phenotypes, with overlapping features [1]. The best-characterized α-synucleinopathies are Parkinson’s disease (PD), Parkinson’s disease dementia (PDD), dementia with Lewy bodies (DLB), and multiple system atrophy (MSA), but α-syn pathology is also seen in neuroaxonal dystrophies [2]. Traditionally, distinguishing α-synucleinopathies has relied on the clinical presentation and post-mortem neuropathology [3]. The neuropathological hallmark comprising α-syn aggregates in neurons, Lewy bodies (LBs), and Lewy neurites (LNs), characterizes PD, PDD, and DLB, while glial cytoplasmic inclusions (GCIs) within oligodendrocytes predominate in MSA. Both neuronal and glial inclusions are present across α-synucleinopathies with disease-specific signatures [4,5].

PD, the most prevalent of these diseases, is characterized by the cardinal clinical signs of tremor, rigidity, bradykinesia, and postural instability, caused by the death of dopaminergic neurons in the substantia nigra. PDD and DLB additionally present with cognitive impairment 1 year after or prior to the motor symptoms, respectively [6]. In PD, neuronal loss is often confined to the brainstem and subcortical regions, while PDD and DLB display neocortical involvement. Neocortical LB involvement is thought to underlie the fluctuating cognition, visual hallucinations, and executive function deficits seen in PDD and DLB. MSA demonstrates fundamental clinical differences, with varying combinations of autonomic impairments, cerebellar ataxia, and parkinsonism, along with a shorter life expectancy. Recent studies suggest that pure autonomic failure (PAF) and rapid eye movement sleep behavior disorder (RBD) represent early disease phenotypes of α-synucleinopathies, with phenoconversion rates as high as 6.25% to 14% annually [7,8].

α-Syn is a 14 kDa cytoplasmic protein encoded by the *SNCA* locus on chromosome 4. It is thought to be involved in the formation of SNARE protein complexes (the vesicle-fusion machinery) and neurotransmitter regulation, as it is abundant at the presynaptic terminals [9]. While classically described as intrinsically disordered, unfolded monomers have been proposed to exist, under physiological conditions, in a dynamic equilibrium with α-helically folded tetramers, which are detected using a cross-linking assay, and show a low aggregation propensity [10]. Additionally, α-syn can adopt a variety of conformations and multimerization states, including beta-sheet-rich oligomers and fibrils [11]. However, the exact mechanisms leading to the aggregation of α-syn under pathological conditions remain elusive. The ability of α-syn to induce the misfolding of α-syn monomers into aggregates in a prion-like manner allows for the propagation of misfolded α-syn, and is termed ‘seeding’ [12]. The accumulation of α-syn multimers in pre-synaptic terminals has been shown to impede neurotransmitter release and synaptic function, and ultimately results in a dying-back-like phenomenon of axonal loss [13].

Missense mutations in the amino terminus of *SNCA*, including A53T, A30P, E46K, G51D, and A53E, along with copy number variations, such as duplications and triplications, yield autosomal dominant diseases with similar phenotypes, and a clear cumulative dosage effect. Almost absolute penetrance has been associated with triplications and certain missense mutations [14]. No familial mutations causing MSA have been confirmed, although the missense mutations G51D and A53E in *SNCA* can give rise to atypical mixed PD and MSA pathologies [15]. Thus, these *SNCA* mutations have provided new insights into the interlinked pathophysiology of α-synucleinopathies, and a possible mechanism of molecular dysfunction. The autosomal dominant mode of inheritance, and the high penetrance of most of these mutations insinuate a gain in function mechanism in familial PD linked to *SNCA* mutations, while the clinicopathological similarities suggest that these mechanisms could also be involved in idiopathic disease. An important distinction to note is that, presumably, in these familial PD types, the chain of events leading to α-syn prion-like aggregation and neuronal death is triggered by a genetic mutation, whereas, in sporadic disease, the initial trigger and perpetuating mechanism remain unknown. They presumably involve a combination of environmental and endogenous genetic factors, leading, as in familial PD, to an increased proportion of oligomers and fibrils.

Alternatively, it has been proposed that these disease-causing mutations could exhibit somatic mosaicism, where only the cells that acquired the mutation can seed, but the evidence to support this in PD is lacking [16]. Mutations alter the way in which α-syn interacts with itself, perturbing the proportion of α-syn species, by favoring oligomers and fibrils [17]. Despite the established role that α-syn plays in familial PD cases, the simple overexpression of α-syn does not recreate the full clinicopathological phenotype of α-synucleinopathies in vivo. The first overexpressing model of wild-type (WT) α-syn was driven by the platelet-derived growth factor-β promoter, and developed progressive nuclear and cytoplasmic neuronal inclusions in the cortex, hippocampus, and substantia nigra. These inclusions were associated with decreased striatal dopamine and with motor deficits at ~12 months [18]. However, other studies expressing WT or mutant α-syn under the control of different promoters (Thy-1 and tyrosine hydroxylase), despite showing inclusions, fail to reproduce the motor phenotype or dopaminergic deficits [19,20]. Therefore, although the events that lead to the assembly of soluble α-syn into insoluble filamentous assemblies constitute the seminal pathological event in these diseases, additional factors in α-syn aggregation may be required for the development of disease.

### 1.2. The Prion Hypothesis of α-Syn and the Introduction of ‘Strains’

The prion hypothesis of α-syn explains how misfolded α-syn templates the misfolding of endogenous α-syn, and disseminates throughout the brain. This disease mechanism was first discovered in a group of transmissible neurodegenerative prion disorders, including Creutzfeldt–Jakob disease (CJD) and scrapie [21]. While the majority of CJD cases are sporadic or inherited, less than 1% of cases are acquired via transmission between individuals. In contrast with human prion disease, the transmission and infectivity of PD between individuals has not been demonstrated. The prion-like spread of α-syn was first suggested after the observation of host–graft transmission visualized via the presence of LBs in grafted fetal neurons within the substantia nigra of PD patients [22,23]. It is uncertain how the α-syn aggregates appeared in these unrelated donor cells, but it has been shown that α-syn can be released from both neurons and glia and, subsequently, taken up by both [24]. The α-syn pathology in PD also follows a stereotypical pattern of neuroanatomical distribution reflected in the Braak staging scheme, supporting the idea that neuroanatomical pathways act as conduits for disease spread [25]. Since then, the ‘prion-like’ propagation of misfolded α-syn has been demonstrated in vitro and in vivo, using both recombinant and endogenous proteins. TgM83 mice express mutant human α-syn under the control of the prion-protein promoter and, when asymptomatic mice are inoculated with brain tissue from symptomatic mice, they display an accelerated onset of α-syn aggregation, neurological symptoms, and death. Interestingly, the control mice do not display these features following inoculation [26]. This accelerated disease course could be consistent with the ‘prion-like’ propagation of the disease within a susceptible individual.

One contested point is the explanation of how a single protein, such as α-syn, could cause a spectrum of diseases. The concept of ‘strains’ proposes that the morphological, pathological, and functional properties, as well as the disease phenotype, are determined by the distinct conformation into which a protein misfolds [27]. The properties of two distinct prion strains were shown to be maintained upon a serial passage in animals, and, thus, are likely to be enciphered in the conformation of the misfolded prion protein [28]. Akin to the prion protein, it has been proposed that α-syn forms different strains with different properties, which may underlie the differences seen across α-synucleinopathies [29]. Previous studies suggest that this concept also extends to other neurodegenerative proteins, such as amyloid-beta (Aβ), tau, TAR DNA binding protein 43 (TDP-43), and Huntingtin, as well as the diseases with which they are associated [30,31,32,33]. The question of whether prion-like self-templating is always pathological remains controversial, as there are some beneficial proteins with this ability. Prions found in yeast and drosophila have been shown to implement protein-based epigenetic inheritance and memory persistence [34]. However, the self-templating of α-syn displays key differences to these physiological prion-like systems; chiefly, the lack of regulatory control limiting its dissemination, which supports its role as a pathogenic agent [35]. Therefore, the α-syn present in α-synucleinopathies displays prion-like properties, without retaining all the characteristics of prion diseases [36].

## 2. Molecular Diversity of Exogenous and Endogenous Alpha-Synuclein

### 2.1. Molecular Diversity of Exogenous Alpha-Synuclein

In a protein, amino acids within a polypeptide chain are not static, and are able to rotate at various angles relative to the backbone. This allows them to adopt multiple conformations or folding patterns. α-syn is no exception, and displays a remarkable conformational plasticity, partly due to the presence of an intrinsically disordered region, which lacks a fixed three-dimensional (3D) structure [37]. Both preparations of α-syn species made with exogenous synthesized recombinant protein, and α-syn species obtained from a diseased human brain, display molecular diversity. Accumulating evidence has shown that recombinant α-syn monomers can form aggregates with distinct conformations and properties in different external conditions [38]. This discovery of the in vitro formation of α-syn fibrils with distinct conformations provided fundamental evidence for the strain hypothesis. ‘Strains’ display functional differences associated with the conformational diversity of α-syn amongst the α-synucleinopathies, including, but not limited to, the ability to seed, disseminate, and interact with cells.

Bousset et al., were the first to generate conformationally distinct strains of aggregated WT α-syn in vitro, by varying the buffers and salinity [39]. Transmission electron microscopy identified a cylindrical aggregate termed ‘fibrils’, as well as a flat twisting aggregate termed ‘ribbons’, which displayed distinct seeding capacities, toxicity, inclusion formations, and dissemination routes, as well as faithful reproduction upon elongation with monomeric α-syn. Following their injection into the rat brain, ribbons induced the formation of α-syn aggregates in oligodendrocytes in a shorter time span, reminiscent of the MSA-like phenotype, while the injection of fibrils was more cytotoxic, and resulted in a pronounced progressive motor impairment [40,41]. These same polymorphs display differential binding and clustering at the neuronal membrane, with distinct patterns of synaptic receptor redistribution [42]. However, considering that ‘ribbons’ are generated under non-salt conditions, the lack of such an environment in the human brain casts doubt on their existence in vivo. Apart from the presence or absence of salt, other factors, such as endotoxins, can also influence the structure and seeding properties of α-syn fibrils [43]. Physical features, such as the bending rigidity and persistence length, have been observed to vary during the generation of structurally and functionally distinct polymorphs with high homogeneity under stringent experimental conditions [44]. An increased protein flexibility is thought to affect ligand binding, which could perturb α-syn function; namely, elongation, binding interactions, or cell internalization [45,46]. The discovery of strain-specific pathology and behavioral and functional traits with exogenous recombinant α-syn is fascinating, especially when extrapolated to a human in vivo setting, providing insights into how endogenous pathological α-syn can give rise to various α-synucleinopathies.

### 2.2. Molecular Diversity of Endogenous Alpha-Synuclein Present in Postmortem Human Brain Tissue

There is evidence demonstrating the presence of different α-syn conformations, or ‘polymorphs’, of endogenous protein from the postmortem brain from α-synucleinopathies, akin to those described with recombinant α-syn. However, it is important to note that recombinant α-syn fibrils formed in vitro do not replicate all the characteristics of the amplified α-syn derived from the brains of α-synucleinopathy patients. They display more variability amongst strains, and are less robust at inducing disease, suggesting that other factors apart from the α-syn conformation contribute to pathogenesis. The ubiquitous ability of recombinant α-syn fibrils to infect all cell lines is also not observed with endogenous α-syn, suggesting that recombinant α-syn fibrils are not completely representative of disease-causing strains. In postmortem brains, Spillantini et al. were the first to demonstrate that α-syn filaments from MSA can display straight or twisted conformations, with reported diameters of 10–30 nm, while PD/DLB filaments display 5 nm protofilaments, two of which can associate to produce a variably twisted filament [47,48]. Furthermore, amino-terminus-specific anti-α-syn antibodies consistently label one filament end, suggesting that the epitope is hidden within the body of the filament and exposed only at one end, giving α-syn filaments a degree of polarity. In line with this, the in vitro α-syn conformations described by Bousset et al. are often likened to the α-syn derived from MSA and PD brain tissue, which largely resembles the ‘ribbon-like’ polymorph, while the DLB strains bare more structural similarities to the ‘fibril’ polymorph. These differences are retained after injection into the rat brain in vivo, with MSA- and PD-derived α-syn inducing the neurological deficits and the dissemination of the α-syn pathology, while DLB-derived α-syn fails to induce any overt neuropathology in this model [49].

In the following sections, we will focus on the studies that have discerned the biochemical, structural, toxicity, seeding, and spreading patterns of endogenous α-syn obtained from post-mortem tissue from patients with α-synucleinopathies. These properties seem to underlie the distinct phenotypes observed across conditions; therefore, understanding the correlation between α-syn strains and the properties conferring these characteristics is key.

#### 2.2.1. Diverse Sensitivities to Proteases and Detergents

The different structures of prion strains were illustrated by the presence of diverse proteolytic fragments, following enzymatic digestion. Proteinase K (PK) and thermolysin digest proteins based on the exposure of cleavage recognition sites within the 3D structure of the protein. Studies using patient-derived α-syn also demonstrate the presence of these diverse proteolytic fragments and, thus, given that the sequence of amino acids remains the same, indicate distinct 3D structures [50]. This resistance to proteolysis has been investigated with both the end products of seeding assays, and the original α-syn fibrils, to determine if this is a property that can be faithfully maintained during amplification [49].

When exposed to PK digestion, LBs yield smaller cleaved fragments, while GCIs are largely undigested [51]. This finding is not limited to the brain, as amplified MSA α-syn products from the olfactory mucosa were also more resistant to PK digestion than PD samples [52]. While an increased PK resistance of GCIs could correspond to a reduction in the degradation rate of GCIs compared to LBs, which could promote the faster spreading of the α-syn pathology in MSA, this property may be protease-dependent. For example, the trypsin proteolysis of sarkosyl-insoluble fractions from GCIs yielded more bands than LBs, while thermolysin digestion did not [51]. To add to the complexity of inclusion protease resistance, after endogenous PD- and MSA-derived α-syn are amplified via a seeding amplification assay, they display similar limited PK-degradation patterns, resembling that of the polymorph ‘ribbons’. Contrastingly, the structure of DLB fibrils was similar to the polymorph ‘fibrils’, and conformationally distinguishable from MSA/PD fibrils using conformational antibodies [49]. This suggests that fibrils derived from one α-synucleinopathy possess a unique biochemical ‘signature’, with PD and MSA resembling each other more than DLB-derived fibrils. Considering that LBs predominate in PD and DLB, it is interesting that they do not have more in common, according to these assays. This highlights other factors that may influence aggregation. The amplified α-syn fibrils of patients with PD or MSA were nevertheless distinguished when amplified from the cerebrospinal fluid (CSF), displaying a high PK resistance, but with a consistently different proportion of protease-resistant bands mapped to the middle region of α-syn, supporting the one-strain–one-disease hypothesis. Two additional heavier bands were seen in PD compared to MSA, and this signature was, overall, maintained after seeding; there was a small variability [29]. Thus, whether α-syn seeding assays faithfully maintain biochemical and structural properties remains disputable, and further work is needed, to understand why, in some cases, faithful templating does not occur. This may be technical, due to differences in protocols, or a change in the α-syn biochemical properties caused by the amplification itself.

α-syn obtained from MSA samples has been shown to be less conformationally stable following exposure to detergents. Guanidine hydrochloride, a chaotropic agent that denatures using favorable interactions with the polar parts of proteins, showed that MSA samples were less stable than PD [53]. Similarly, treatment with sodium dodecyl sulfate (SDS) yielded more insoluble α-syn in DLB and PD, compared to MSA [54]. An increased resistance to detergents implies a higher thermodynamic stability, or less polar interactions within the protein, due to tight packing. This may mean that MSA aggregates are smaller and more easily fragmented by detergent-like molecules, increasing their propagation propensity through being less stable [55]. This might also explain the faster disease progression seen in MSA, and raises the question of whether pathogenesis may be independent from the insoluble α-syn aggregates.

One important caveat to note is that resistance to proteolytic digestion and detergents comprises blunt tools that can be affected by many factors that can all modify access to the different parts of α-syn species (i.e., the primary amino acid sequence, 3D quaternary structure, isoforms, and post-translational modifications) and, therefore, further evidence is needed, to fully understand these effects. However, despite their limitations, these assays provide additional evidence of the inherent conformation differences amongst the α-syn “strains” associated with α-synucleinopathies.

#### 2.2.2. Ultrastructure Diversity

The differing resistances of α-syn varieties to protease digestion and detergent treatment support the idea of different 3D arrangements (secondary, tertiary, and quaternary structures), which could account for the clinical–pathological differences seen across α-synucleinopathies. This concept has been investigated with prion strains using cryo-electron microscopy (cryo-EM) and computational 3D reconstruction, which can provide a resolution at the amino-acid level [56]. Recent cryo-EM studies have provided further evidence for the different structural architectures of PD, DLB, and MSA α-syn aggregates [57,58,59].

The idea of distinct conformations and replication mechanisms underlying MSA and PD is highly supported by the finding that, although using WT α-syn as a substrate facilitates seeding with PD and MSA brain tissue, α-syn with the E46K mutation inhibits MSA-derived, but not PD-derived, strain replication, in vitro and in vivo [60,61,62]. α-syn fibrils in MSA patients consist of type I and II filaments that co-exist at different ratios amongst patients, and are each composed of two dissimilar protofilaments packed asymmetrically (Figure 1). Furthermore, the core of the larger protofilament of the type I filaments contains a different sequence of residues compared to the other protofilament, as well as both the protofilament cores of type II filaments. Crucially, they all contain an intramolecular salt bridge between residues E46 and K80, which stabilizes a Greek key motif, a form of parallel in-register beta-sheet architecture [57]. Therefore, without the E46 residue, the MSA α-syn fibrils cannot template their structure, but PD and DLB fibrils can, indicating a different binding interface and, by extension, a different structure and misfolding mechanism. The cryo-EM structures of recombinant E46K α-syn fibrils lack this salt bridge and, instead, adopt an alternate conformation, stabilized by a salt bridge, between residues K45 and E57 [63]. The E46K mutation is known to disrupt one of the *KTKEGV* repeats, possibly impeding the formation of the original salt bridge, due to the unfavorable electrostatic repulsion between the positively charged K46 and K80 and, therefore, may influence α-syn strain propagation, akin to natural selection. Intriguingly, the protofilaments also form a cavity encompassing an additional density of unknown origin, which indicates the presence of a non-proteinaceous, yet unidentified, molecule that could have functional consequences [57].

On the other hand, filaments isolated from PD and DLB patient brains are thinner, less twisted, and are composed of a single α-syn protofilament (Figure 1) [57]. These fibrils display an alternative structure, with a salt bridge between E35 and K80, and an ordered core termed the “Lewy fold” [64]. This suggests that the E46K mutation, rather than disrupting the internal structure of α-syn, instead likely alters the protein misfolding substantiated by successful protein propagation with PD/DLB-derived fibrils. The 3D quaternary structure of α-syn fibrils has been shown to be a critical determining factor in the seeding of pathology, and the ability to recruit endogenous α-syn monomers [59]. It remains to be discovered if this is due to the differences discovered in the filament interfaces, the helical arrangement, or other, unknown features. The study also established, with a maximum resolution of 2.2 Å, that filaments with the Lewy fold have a right-handed twist, in contrast to the left-handed twist observed for the MSA-fold α-syn filaments [64]. This distinct handedness demonstrates that α-syn fibrils can be polymorphic at the molecular level, but we can only speculate the effect that this has on the protein function. Enzymes are known to exhibit protein–substrate stereoselectivity; thus, demonstrating an alternate chirality may inhibit the enzymatic degradation of α-syn via the autophagy or ubiquitin-proteasome pathway [65]. Alternatively, it could create a barrier to seeding, where chirality must be conserved in order to seed efficiently, as seen with Aβ [66]. It raises questions about the origin of this helical chirality in the fibrils, as well as if and how it modulates pathogenicity. A recent study demonstrated that patient-derived Aβ fibrils display a right-hand twist, contrary to the left-hand twist seen in the in-vitro-formed Aβ fibrils [67]. This highlights the need to validate that this is not an artificial change occurring from the seeding process and that it is, thus, unlikely to occur endogenously in the brain.

Ultrastructural evidence, thus, further supports distinct α-syn aggregate structures being present in MSA and PD/DLB, lending credence to the notion that α-synucleinopathies are caused by distinct α-syn strains. Interestingly, the structural commonalities of PD and DLB α-syn filaments suggest that a unifying α-syn strain may underlie them and, thus, other factors, such as differences in the spatial and temporal emergence, may account for the clinical and pathological heterogeneity observed. Therefore, this suggests that there is a combination of factors, both intrinsic and extrinsic to α-syn, accounting for the clinical–pathological differences between α-synucleinopathies.

#### 2.2.3. Diverse Seeding Propensities

Cell-free seeding amplification assays (SAAs), such as the real-time quaking-induced conversion (RT-QuIC) and protein-misfolding cyclic amplification (PMCA) assay exploit the ability of α-syn to induce monomers to misfold in a cyclical fashion, and aggregate into α-syn fibrils (Figure 2). During the formation of these aggregates, the α-syn also incorporates a fluorescent dye, thioflavin T (ThT). This increase in fluorescence acts as a proxy measure for aggregation in real-time. This assay was originally developed for the detection of prions in CJD samples, but has been repurposed for the detection of proteins with prion-like properties, such as α-syn and tau [68,69,70,71]. This repurposed version has been used to investigate differences in α-syn obtained from different α-synucleinopathies, as a possible clinical diagnostic tool. In this section, we will discuss the use of SAAs in investigating the seeding kinetics of α-syn from α-synucleinopathies, and whether this evidence supports the existence of multiple α-syn strains.

Fairfoul et al. were the first to use the RT-QuIC assay to detect α-syn aggregation within the brain tissue and in CSF [69]. Since then, SAAs have been optimized to detect α-syn in saliva, skin, and the olfactory mucosa. Studies have used them to investigate the seeding kinetics of PD, MSA, and DLB, through comparing kinetic parameters, such as the lag phase, maximum ThT fluorescence, time to maximum fluorescence, protein aggregation rate, and area under the curve. Postmortem MSA brain tissue and CSF have been shown to induce faster aggregation kinetics than PD-derived samples. However, despite aggregating faster, MSA-derived samples reach a lower fluorescence plateau than PD [29]. This increased propensity to aggregate might reflect the more aggressive nature of MSA; however, it is the lower fluorescence plateau that could provide information about the structure. ThT is known to bind to beta-sheet surfaces, secondary structures present in α-syn fibrils [72]. The lower fluorescence plateau represents a stage when all the ThT molecules have bound but, as this parameter is not constant, it suggests that there is another rate-limiting factor, likely the exhaustion of the α-syn monomers initially present. Therefore, a lower fluorescence plateau could mean fewer beta-sheets in the MSA-derived protofilaments. This is supported by recent cryo-EM-resolved protofilament folds of the PD- and MSA SAA-end products being highly similar, but the PD fold was composed of eight beta-sheets, whilst the MSA fold only contained seven beta-sheets [59]. However, different results have been obtained with samples originating outside the central nervous system (CNS). The opposite effect was observed using salivary samples, with PD samples aggregating faster, while cutaneous PD and MSA samples show similar kinetics [73,74]. Clearly, further work is needed, to understand if and how the anatomical location, particularly within the CNS versus outside the CNS, can influence the strain configuration.

Differences in α-syn aggregation kinetics have also been observed between DLB and PD samples, using both brain samples and antemortem CSF. Samples from patients who lived with DLB have been shown to have a higher fluorescence maximum and faster aggregation kinetics compared to PD samples [75,76]. Furthermore, a 1000-fold dilution with those DLB samples was sufficient for a positive SAA, unlike MSA and PD, suggesting an increased α-syn seeding potency in the DLB samples. However, increasing the CSF concentration does not seem to have a linear effect, with the amplification signal plateauing. This suggests the presence of a putative inhibitory mechanism present in CSF, which is yet to be identified, and highlights that we do not understand the seeding mechanism entirely. This putative inhibitory mechanism may be due to other molecules present at low concentrations, or possibly α-syn itself. The phosphorylation of α-syn at Serine 129 (S129) is the most common post-translational modification that has been reported to inhibit α-syn aggregation [77,78]. This might explain, at least partly, how increasing the sample concentrations results in a seeding plateau, as the phosphorylated α-syn may reach the minimum concentration needed to produce an inhibitory effect.

The high sensitivity and specificity of SAAs have enabled them to discriminate between α-synucleinopathies, non-α-synucleinopathies, and healthy controls [79]. Other studies have found that RBD and PAF, which are thought to represent the prodromal stage of α-synucleinopathies, are able to yield positive α-syn SAA results, with a sensitivity of 95.3% [80]. Patients with RBD and a high risk of phenoconversion have also been shown to display positive seeding responses [69]. This raises the question of whether seeding-competent α-syn is already widespread, prior to detection with conventional means, such as immunohistochemistry. It also highlights whether, at one point, there is a commitment to a specific disease strain or clinical phenotype. Further work is needed, as a paucity of studies exists investigating whether people with RBD and PAF with a positive SAA display different kinetic parameters, depending on to which of the α-synucleinopathies they phenoconvert.

Although the demonstrated sensitivity and specificity of SAAs to discriminate across α-synucleinopathies are encouraging, they remain compounded by a co-existing or restricted spread of pathology. The RT-QuIC assay sensitivity is limited when samples are derived from cases with brainstem-only pathology, as well as Alzheimer’s disease (AD) with LBs [69,79]. These results raise the questions of what the full extent is of the molecular diversity in α-syn strains, whether it changes over time, and, if incidental LBs are composed of the same α-syn strain, why they do not always progress into full-blown PD. Methodical variabilities, such as the incubation times, temperature, and buffers may contribute to the variability in the results. These protocol variations may select for different conformations and multimerization states of α-syn, which will likely display different properties. In addition, physiological factors may not be permissive for seeding certain strains of α-syn. Moreover, it is possible that the optimal conditions to amplify the strain associated with MSA are distinct to those for the strain associated with PD. A combination of these reasons may explain why some studies found only 7–35% of MSA CSF samples, but other studies found the totality of PD samples, were capable of seeding samples [80,81,82].

These studies illustrate that SAAs comprise a powerful tool to detect α-syn seeding, in both a lab and a clinical setting. The variable responses amongst the α-synucleinopathies are encouraging, and strengthen the idea that multiple strains may exist; however, future studies are needed, to establish a standardized protocol.

#### 2.2.4. Diverse Abilities to Exert Inclusion Formation and Toxic Effects on Cultured Cells

Cellular exposure to α-syn aggregates in vitro results in a mounted cellular response, as well as cytotoxicity. In this context, cell models have been used to explore how toxic events are modulated by different α-syn strains on various treated cell types. There are engineered in vitro systems that allow for the tracking of the intracellular aggregation of α-syn with a stable expression [77]. The focus of this section will be on expanding on how in vitro models have been used to explore the differential cell vulnerability and variable inclusion morphology associated with α-synucleinopathies.

α-syn obtained from the diseased brains of α-synucleinopathies can form various diverse intracellular α-syn aggregates in cultured cells. Biosensor HEK293T cells stably express human α-syn with the A53T mutation, developing intracellular α-syn inclusions upon exposure to α-syn. These inclusions were seen after treatment with α-syn obtained from both the detergent-soluble and -insoluble MSA brain tissue fractions, while PD samples only showed seeding when the detergent-insoluble fraction was used [83]. This suggests that the detergent-insoluble fraction, presumably enriched in aggregates, displays an ability for seeding in vitro, but the fact that the seeding ability of MSA α-syn was not limited to the insoluble fraction may be explained by MSA α-syn being more vulnerable to the detergent denaturation process, which could create smaller oligomeric complexes, more suited for propagation [55]. Similarly, PD α-syn has a limited ability to propagate in biosensor cells, while the ability of α-syn from DLB samples to propagate was increased following PK digestion and phosphotungstate anion precipitation, a technique used to precipitate prions [84]. Intracellular α-syn inclusions induced in vitro following treatment with α-syn have been shown to also differ morphologically. The exposure of HEK293T-α-syn-(A53T) cells to MSA samples results in thread-like, cytoplasmic inclusions, while PD samples induce smaller and circular inclusions. The cellular localization of GCIs resembles the cytoplasmic inclusions induced in these biosensor cells, whereas the circularity of the inclusions induced with PD samples resembles LBs [83]. In another model, the exposure of primary cortical neurons with SAA-amplified fibrils resulted in thick linear neurites with a punctate pattern of α-syn inclusions upon exposure to MSA and PD samples, while DLB samples yielded diffuse somatic inclusions around the nucleus [49]. As MSA and PD α-syn are suggested to be ‘ribbon-like’, and DLB α-syn is ‘fibril-like’, these differences may be due to ribbons requiring a longer incubation period to induce pathology. Therefore, the α-syn from MSA and PD samples may display a reduced efficiency, creating α-syn inclusions in neurites [39].

Differences in the cytotoxicity elicited by different species of α-syn within cultured cells have also been studied. The treatment of HEK293A cells with recombinant α-syn induces cytotoxicity, which correlates with an increasing concentration of α-syn. This linear relationship between the cytotoxicity and α-syn aggregate concentration was retained with α-syn from PD and DLB temporal cortices, although the gradient differed. The highest amount of cell death occurred when aggregates; ~450 nm were the dominant species, with monomers and fibrils inducing a smaller cytotoxic response [85]. This shows that there must be inherent differences in the composition of the α-syn species found in PD and DLB, which might affect the aggregation rate and cellular interaction ability. The importance of size for α-syn aggregates could also contribute to the variability in SAA seeding responses seen previously [85]. The treatment of neuron-like SH-SY5Y cells with SAA-seeded MSA olfactory mucosa showed increased levels of signaling molecules associated with an inflammatory response, compared to the control and PD samples, while no inflammatory response was observed with recombinant α-syn. This study also showed that MSA samples had a higher resistance to PK digestion, thus implying the existence of a structure–function relationship, where morphology influences immunogenicity in α-syn [52].

Altogether, these results demonstrate that the cellular response depends on the α-syn strain added, as well as the cellular localization and morphology of the formed inclusions. MSA-derived α-syn often has a higher cytotoxic effect, which reflects the more aggressive disease phenotype. These findings give some insights into the differences in cell vulnerability observed across α-synucleinopathies, with oligodendrocytes predominately affected in MSA, and neurons in PD, as well as clues to the basis for the morphology of inclusions formed in these conditions.

#### 2.2.5. Diverse Dissemination Patterns In Vivo

Upon in vivo inoculation, α-syn aggregates display dissemination throughout susceptible brain regions, and the development of further pathology. This prion-like spread is observed using both exogenous α-syn and endogenous α-syn obtained from human tissue. The pattern of spread is shown to be dependent on the neuroanatomical location of the inoculation and the time point, as well as the α-syn strain used. In this section, we will review the studies that have explored α-syn spreading in vivo across α-synucleinopathies. Mice have been commonly used, mainly TgM83+/− mice, which are M83-transgenic mice expressing human A53T mutant α-syn under the mouse prion promoter.

Prusiner et al. first demonstrated that the prion-like spread of MSA-derived α-syn after injection into TgM83+/− mice induced neurological dysfunction. The spread could not be replicated with the PD, control [86], DLB or AD samples [87,88]. In contrast, other studies have reported α-syn propagation using detergent-insoluble fractions from PD and DLB samples in mouse models. The injection of the sarkosyl-insoluble fraction from the cortices of MSA and incidental Lewy body disease patients into mice expressing WT human α-syn resulted in the development of a similar spatiotemporal pattern of spread and neuropathology at 6 months post-injection [89]. Similarly, another study successfully propagated DLB-derived α-syn from the sarkosyl-insoluble fraction in WT mice, although recombinant α-syn displayed a higher propagation efficiency [90]. This shows that independent of the in vitro or in vivo setting, MSA-derived α-syn can propagate with a higher efficiency than PD and DLB samples, for which propagation is only achieved using detergent-insoluble samples enriched in α-syn aggregates.

The spread potential of seeds from peripheral tissues has also been assessed. Two studies induced progressive pathology in WT mice and non-human primates with sonicated LB-enriched fractions obtained from the substantia nigra of PD samples, but not from the stellate ganglion [91,92]. In contrast, another study demonstrated subtle α-syn pathology in the somatodendritic compartment and dystrophic neurites in 96% of TgM83+/− mice injected with SAA-competent brain tissue from PD patients, but only in 50% of mice injected with SAA-competent PD stomach wall tissue [93]. This demonstrates the variability in peripheral nervous tissue samples in inducing α-syn pathogenesis in vivo.

These experiments show that postmortem α-syn can spread in a prion-like manner in vivo, in line with in vitro SAA results. MSA-derived α-syn propagates more efficiently, reflective of the more aggressive phenotype of MSA and that the in vivo propagation of PD and DLB samples is usually limited to detergent-insoluble fractions. While this last point could imply that the most pathogenic species of α-syn in vitro and in vivo are the larger insoluble aggregates, most of these studies used sonicated extracts, which may have released small oligomers. Perhaps, factors present in the cellular milieu might limit the propagation ability of PD/DLB soluble α-syn species. Alternatively, the intrinsic properties of soluble PD/DLB oligomers are sufficient to induce α-syn seeding in SAA, but not propagate in vitro and in vivo, while soluble MSA-derived α-syn is able to do both.

A limitation encountered by some of these in vivo studies seems to be difficulty in observing significant neurological dysfunction in the inoculated animals [93]. While this could be due to the limited incubation time, shorter that required to elicit functional consequences, it may also reflect the insufficiency of the pathological outcome measurements used (for example, if limited to phosphoroylated S129 α-syn detection) to reflect symptomatic disease.

In summary, there is a breadth of evidence representing the different properties of α-syn strains and their relationships with the different α-synucleinopathies using endogenous α-syn. While the reason behind these different α-syn conformations in different diseases remains elusive, it is likely to be influenced by other unknown factors extrinsic to the strains themselves, as generated recombinant α-syn strains do not resemble endogenous α-syn obtained from postmortem tissue in in vivo and in vitro experiments.

## 3. Disease Variability

### 3.1. Correlation between α-Syn Strains and Disease Phenotypes

There is a substantial clinical–pathological heterogeneity within α-synucleinopathies. In this review, we propose that at least part of this heterogeneity may be explained by the distinct strains of α-syn between the individuals and neuroanatomical regions involved. This disease heterogeneity is reflected in the structural differences in α-syn, not only between the different diseases, but also within each disease. For example, a recent study found a range of SAA responses within each disease, and this range was broader in PD than DLB [75]. Similarly, cryo-EM has shown that MSA-derived α-syn can adopt at least two structural polymorphs within a single disease, which might further vary between clinical cases [57]. Similarly, it has been demonstrated that amplified α-syn derived from PD samples displayed structural heterogeneity, even more so than in MSA-derived samples [94].

Clinically, MSA can present as either a cerebellar (MSA-C) or parkinsonian (MSA-P) subtype, depending on the predominant symptoms, and the brain regions involved. However, despite the observable clinical distinction, the neuropathology at autopsy is more uniform. It has been shown that MSA-P samples, either from the olfactory mucosa or CSF, induced a positive SAA response, which was not observed with MSA-C samples [75,82]. The lack of seeding in MSA-C suggests that, although MSA-P and MSA-C are subtypes of the same disease, they may be caused by distinct α-syn strains that include different tropisms for peripheral tissues, including the olfactory mucosa. Alternatively, the properties of α-syn may be influenced by the microenvironment, and lead to the onset of different forms of MSA. Further studies are needed, to investigate the seeding potential of different brain regions between MSA-C and MSA-P, and to elucidate if different α-syn strains are present. A regional seeding variability was also observed in PD, and it correlated with the predominant clinical presentation. Hippocampal α-syn from patients with a predominantly cognitive deficit at presentation showed a higher seeding than those with a predominately motor-based presentation, whose nigral α-syn, instead, trended toward a higher seeding. This suggests that the more-involved regions may have a higher α-syn seeding ability. Additionally, the more-affected regions showed a higher resistance to thermolysin digestion, linking back to a possible correlation between the α-syn quaternary structure and a faster disease course. Interestingly, the pons and cerebellum exhibited a low seeding; these regions are often affected early, which suggests that the α-syn seeding capacity may diminish over time, as the oligomers are sequestered into larger insoluble α-syn aggregates [95]. It may also indicate that regions may have a dissociated ability for the replication of seeds and propagation, with some regions behaving as replication cauldrons, while others may behave like propagation motorways [35].

This observed variability is congruent with the idea that α-synucleinopathies may not represent separate disease entities but, rather, a single disease spectrum. Growing evidence suggests that neurodegenerative cases are composed of different combinations of multiple polymorphs, which populate diseased brain tissues, named the ‘cloud hypothesis’ [27]. According to this hypothesis, there is not a single MSA or PD strain but, rather, a combination of α-syn polymorphs, one of which is the ‘most prevalent’, and is able to determine the ‘dominant’ disease phenotype. It also provides an explanation as to why certain seeding assays may select for, and subsequently yield, different polymorphs that are faithfully propagated with findings that are not replicated by other studies. The interrelation between different pathologies and their ability to explain disease variability has been studied in the amygdala. The amygdala in PD was found to contain a higher amount of neuritic pathology, astrocytic pathology, and carboxy-truncated forms of α-syn than that in AD with amygdala-restricted LBs [96]. This suggests that, despite the commonality of α-syn pathology in the amygdala in both diseases, there are intrinsic molecular differences that may result in the predominance of the AD versus PD phenotype. The exact nature of this decisive factor remains enigmatic.

### 3.2. Mechanisms of α-Syn Strain Heterogeneity/Pathogenesis in Alpha-Synucleinopathies

Mounting evidence suggests that distinct α-syn strains are responsible for the diverse pathological presentation of α-synucleinopathies, but how a single misfolded protein can give rise to different disease phenotypes is not defined. There are several mechanisms through which the presence of α-syn strains could result in the generation of a diverse set of phenotypes. Some of these mechanisms might be autonomous to α-syn protein, depending solely on the conformation of α-syn and the amino acid stretches exposed at the extremities. Other mechanisms may also rely on factors external to α-syn, such as the different interactions of α-syn strains with the cellular components of multiple cell types, in a region-specific manner. Understanding the different properties that distinct conformations of α-syn display gives an insight as to the possible mechanism behind the pathogenesis.

#### 3.2.1. Autonomous α-Syn Mechanisms

As previously discussed, the structure of α-syn provides the simplest basis as to how diverse α-syn strains arise. Translating this concept into the distinct pathogenesis observed across α-synucleinopathies, the structure–pathology relationship is further demonstrated by the variable neuronal membrane-binding efficiency displayed across α-syn strain polymorphs [42]. These interactions were shown to differentially re-distribute glutamate receptor subunits on the neuronal membrane and, thus, showed that different strains of α-syn could differentially impact neuronal networks and transmission. Although this study used recombinant α-syn, it is likely that endogenous α-syn may also display these variations. Polymorphic α-syn assemblies amplified in the presence of human PD or MSA brain tissue also exhibit distinct protein interaction patterns, including varying levels of interaction with the PD-associated protein deglycase DJ-1 [97]. The loss of the protective chaperone function of DJ-1 exacerbates α-syn aggregation, by perturbing removal; thus, lower levels of protective proteins offer one possible mechanism behind the observed differences in strain-specific toxicity [98]. This idea is not limited to α-syn, with evidence showing that different conformational strains of tau also have different interactomes, displaying variable associations to crucial cellular proteins [99]. Overall, this highlights how conformation-specific interactions with the cellular proteome could disrupt crucial protein function, likely leading to the differences in toxicity discussed previously, as well as disease characteristics.

Once these polymorphs were internalized by the neurons, they displayed different growth rates [42]. As the polymorphs were fragmented to standardize the length and rate of uptake, it is likely that these polymorphs display different capacities for elongation. The growing ends of the α-syn fibrils likely recruit monomers at different rates, depending on which α-syn conformations are present in the cytosol. This might explain why α-syn from different α-synucleinopathies might result in more aggressive, faster seeding and cytotoxicity, due to the presence of diverse α-syn conformations in the cytosol. Similarly, this has been observed in the SAA responses of CSF from PD and DLB patients with mutations in the PD-causing genes *GBA*, *parkin*, *PINK1*, *DJ1*, and *LRRK2* [100]. These mutations are known to increase the likelihood of developing α-synucleinopathies; thus, it is important to understand how altered protein function might create a predisposition to these conditions. PD patients displayed positive seeding in 93% of patients with severe *GBA* mutations, 78% with *LRRK2* mutations, 59% carrying heterozygous recessive mutations, and none of those with bi-allelic recessive mutations. Those with severe *GBA* mutations showed the highest seeding activity. In DLB patients, all *GBA* mutations exhibited positive seeding. Despite the commonality of α-syn to all these patients, the varied SAA responses suggest that certain mutations affect the function of α-syn, by interacting in an alternative way, which would likely occur through the altering of the 3D conformation of α-syn or binding interactions. These mutations are also associated with differences in clinical course, prognosis, and pathology [101]. Interestingly, PD-causing mutations known to induce conformational changes in α-syn have been shown to also affect function, by reducing fibrilization, increasing oligomer generation, and promoting neuronal toxicity [102,103]. Therefore, this implies that these structural differences between α-syn strains might dictate the interactions between α-syn and other proteins, thus translating into the unique pathological mechanisms observed across conditions. Interestingly, when α-syn assemblies from three MSA patients underwent seeding, the cryo-EM structures showed that the process generated a mixture of filaments, with varying structural equivalences [58]. Therefore, it is unclear how a mechanism of simple elongation could explain the formation of filaments with markedly different protofilament folds, compared to MSA filaments. It raises the question of whether additional factors are required in the templating process, or whether the substrate requires post-translational modifications.

Post-translational modifications are known to occur on α-syn, and have been shown to have functional consequences. The elongation of α-syn phosphorylated at Y39 using a substrate of recombinant WT α-syn gave rise to filaments with a different quaternary structure to that of the original seeds. Structurally, phosphorylation at Y39 incorporates the amino terminus into the fibril core, changes the electrostatic interactions, and allows for the formation of a hydrophilic channel. Upon being added to rat primary cortical neurons, phosphorylated Y39 α-syn fibrils induced more toxicity, and illustrated the possible functional effects of post-translational modifications [104]. The phosphorylation of recombinant α-syn at S129, one of the most common post-translational modifications in α-synucleinopathies, induced a distinct conformation, with increased cytotoxicity compared to WT α-syn [105]. S129 is in the carboxyl terminal region of α-syn, and structural differences in this region have been shown to modulate the proteasome activity and cellular clearance, leading to different degrees of toxic phenotypes [106]. Furthermore, a recent study showed that post-translational modifications in soluble α-syn during the amplification in vitro of α-syn derived from postmortem tissue either promoted or inhibited the propagation of pathology, which was shown to solely rely on α-syn conformation [107]. The role of post-translational modifications in α-syn strain diversity is likely complex and multifaceted. Their association with changes in structure and function provides some insight as to why strains could exhibit different properties. Importantly, it remains unclear at what time point post-translational modifications occur during the process of aggregation, and if this triggers further aggregation, or if it is a result of α-syn aggregation itself. Thus, the effect of single and cumulative post-translational modifications on the mechanism of α-syn aggregation, propagation, and cellular interaction needs to be further investigated. However, evidence suggests that they might dictate, at least partly, differential seeding and pathogenesis observed across brain regions and conditions.

#### 3.2.2. Non-Autonomous α-Syn Mechanisms

Other factors besides the conformation of α-syn could be responsible for the differences observed across α-synucleinopathies. It is important to consider that, although these are not factors directly relying on α-syn conformation, it is likely that these strain-specific α-syn differences could translate into clinicopathological differences.

The cellular milieus of neurons and oligodendrocytes display differences that are reflected in the inclusion structures formed, LBs and GCIs. In addition to the morphological differences in the α-syn inclusions, they also differ with respect to their aggregation properties. It has been shown that amplified MSA fibrils more robustly recruit endogenous α-syn, and evoke a redistribution of the tubulin polymerization-promoting protein TPPP/p25α, compared to the amplified PD fibrils, in mice primary oligodendroglial cultures [59]. Despite structural similarities, other features from the MSA seeds could have been imprinted on the amplified fibrils, allowing them to elicit a more MSA-like response, with an increased propensity to recruit endogenous α-syn and interact with other proteins. This has been also observed in vivo as, upon the injection of GCI α-syn and LB α-syn into mice with oligodendrocyte-specific α-syn expression, the GCI-injected group developed more oligodendrocytic pathology, prior to the LB group. Lower levels of neuronal α-syn pathology were also observed in the LB-injected group. Furthermore, the injection of pre-formed fibrils generated a GCI-like strain with comparable potency, likely mediated by cell factors specific to oligodendrocytes. Contrastingly, when the GCI α-syn was added to primary mouse neurons, they retained GCI-α-syn specific properties, and did not convert to an LB-like strain [51]. This suggests that the generation of a GCI-α-syn strain relies on specific extrinsic ‘factors’ present in oligodendrocytes and, once α-syn is converted into a GCI-like strain, it retains aggressive properties. These factors remain elusive, but it is clear that the cellular milieu can exert a form of ‘selective pressure’, influencing α-syn strain propagation, which further confirms that the microenvironment plays a role in dictating α-syn strain properties or even phenotypes.

Despite the clear involvement of oligodendrocytes suggested by Peng et al., it remains unclear how α-syn ends up in oligodendrocytes in MSA. Both PD and MSA brain tissue have been shown to display overlapping sarkosyl-insoluble proteomes, consisting predominately of mitochondrial and neuronal synaptic proteins [108]. These results support the idea that preassembled building blocks, originating in neurons, contribute to the formation of GCIs in MSA, as well as LBs in PD and DLB, implying a shared neuronal origin amongst α-synucleinopathies. This view would agree with the fact that, in transgenic models lacking an engineered oligodendrocyte-specific α-syn expression, such as the M83+/− mice, there is a paucity of α-syn pathology within oligodendrocytes upon inoculation with MSA lysates [109], meaning that, while neurons express α-syn, oligodendrocytes may not even produce it physiologically [110,111,112]. It is therefore thought that oligodendrocytes might somehow uptake α-syn from the environment, or from neighboring neurons in MSA [113]. The transmission of pathological neuronal α-syn to oligodendrocytes might be reliant on α-syn conformation, given that the internalization of α-syn has been shown to display cell-type-specific differences, dependent on conformation and solubility [114]. Alternatively, unknown factors that are present specifically in the oligodendrocytes of MSA patients may make them more susceptible. Once pathogenic α-syn is within oligodendrocytes, unknown cellular molecules confer an MSA-like α-syn strain, with the ability to continue propagating between oligodendrocytes at a higher potency. This conversion to a more potent strain, allowing propagation at a faster rate, would explain the faster disease progression and more aggressive phenotype seen in MSA. This hypothesis, however, is compounded by the limited presence of neuronal pathology in MSA, which does not always correlate with the burden of GCIs [4]. Given that neuronal pathology is readily detected in vivo following α-syn injection, instead of oligodendrocytic α-syn pathology, as mentioned above, it is possible that, in human disease, α-syn pathology may be present in neurons in MSA at levels not readily detected by immunohistochemistry. The neuronal origin of α-syn raises the question of how its presence in oligodendrocytes converts it to an MSA-like α-syn strain. This could be possibly explained by the induction of a structural change in α-syn that may be irreversible, but it remains elusive what molecules are responsible. It is possible that the mere presence of α-syn within oligodendrocytes is enough to commit to an MSA-like disease phenotype. Finally, the question of whether neurons transmit α-syn monomers or aggregates to oligodendrocytes also requires further investigation.

The SAA responses of α-syn obtained from brain tissue can be modulated by physiochemical factors, such as the pH, salinity, and other features of the environmental milieu that can modulate which seeds ‘thrive’. These factors can be adjusted to favor the seeding of MSA [53]. If different cells and regions differ in these physiochemical factors, then the regional variability seen in seeding may be partly due to these physiochemical differences across brain regions. There is a possible presence of multiple strains, where one ‘dominant’ strain, when forced to amplify in favorable conditions, dominates the population in each region, as described in the “cloud hypothesis”. In a study involving a Contursi kindred patient with familial heterozygous α-syn A53T PD, it was shown that inclusions contained both A53T mutant α-syn and WT α-syn [96]. Thus, this suggests that both α-syn species were present at the same time, and that the aggregation of WT α-syn can be engendered due to the presence of a more aggregation-prone ‘strain’ of α-syn, presumably in a prion-like fashion.

The influence exerted by the microenvironment is supported by the finding that in vitro amplified aggregates are less robust in inducing pathology, compared to amplified patient-derived brain tissue, when injected into mice [49]. The patient-derived strains also retained characteristics of the original disease, with MSA-derived α-syn generating the most aggressive phenotype in terms of inducing motor deficits, nigrostriatal neurodegeneration, α-syn spreading, and inflammation, while DLB-α-syn displayed the mildest phenotype. A similar finding of strain-specific pathology in distinct cell types and brain regions, as well as different rates of network propagation, is also seen in the inoculation of transgenic mice with multiple disease-specific tau strains [115,116]. This highlights how the heterogeneous nature of brain homogenates which contain other proteins or inflammatory components can influence or exacerbate the behavior of α-syn. It also exemplifies the retention of strain-specific disease characteristics, likely through a synergistic effect of α-syn as well as the other molecules present in brain homogenates.

SAA-amplified samples have also been shown to display differences in secondary structure and helical twist periodicity, compared to the originals [29]. It is not definitive whether α-syn conformation is always retained after SAAs, and what determines whether the conformation changes during amplification. However, the intrinsic α-syn property differences between α-synucleinopathies are retained at least partially. We cannot exclude the possibility that the SAA is not identical to the amplification process that occurs within the brain, and that SAA selects for certain strains or characteristics. This structural heterogeneity could influence the binding interactions with other proteins, and the efficacy with which they occur. In turn, the external environment and other molecules could also influence the α-syn strain present. The temporal succession of events remains elusive, although it is likely that the pathogenesis mechanism depends on certain molecules being present, but their nature remains unknown.

### 3.3. Differential α-Syn Seeding Profiles Result in Clinical Phenotypic Diversity

We have established that there are different seeding kinetics in PD, DLB, and MSA tissue; however, how this translates to α-syn strains producing distinct clinical phenotypes remains unclear, as correlation between α-syn seeding properties and disease variability may be reflected differently in each condition. SAA responses seem to correlate with clinical parameters, and have been reported in prodromal stages, so they may represent an early mechanism of the disease process. We will discuss this further in this section.

#### 3.3.1. Correlation with Clinical Heterogeneity

Several studies have investigated how the seeding activity correlates with clinical features, such as the disease duration and course. Half-life values obtained from SAAs have been shown to negatively correlate with the Hoehn & Yahr symptom stage in PD samples, suggesting that increased disability is associated with increased seeding [29]. This is in line with the finding that elevated α-syn fibril-templating activity in the CSF corresponds to a reduced survival in DLB. These properties were also replicated upon the exposure of primary hippocampal neurons to CSF from DLB, with a low α-syn fibril-templating activity, which struggled to form new inclusions, while the high fibril-templating activity induced new inclusions robustly. Curiously, the same study using SAA identified an early abundance of fibril-forming activity from the brain tissue of ~1-month transgenic mice expressing WT α-syn prior to the age-dependent aggregation that occurs at ~1 year [117]. Thus, it is possible that α-syn oligomers, present before cellular inclusions are established, might be contributing, at least partly, to this increased fibril activity. Interestingly, α-syn from patients with rapid disease progression had the highest nigral seeding capacity and, when added to human dopaminergic neuron cultures, induced a higher cytotoxicity [95]. This raises important questions as to whether clinical symptomatology is, at least in part, caused by aggressive region-specific seeding, as well as the mechanism behind how high seeding translates into an aggressive disease course. This correlation between clinical parameters and SAA response has been further described with rigidity and postural instability [75]. This regional variability is supported by temporal and frontal cortex tissue from PD and DLB only showing increased seeding in PD-derived frontal cortex samples, when using C-terminally truncated α-syn as a substrate [118].

#### 3.3.2. Correlation with Clinical Heterogeneity in the Prodromal Phase

Currently, it is thought that oligomers mediate α-syn seeding so it is interesting that positive seeding has been seen in PAF and RBD, possible prodromal forms of α-synucleinopathies. A longitudinal study followed patients with RBD, and found 62% converted to PD or DLB during follow-up, of whom 97% were CSF SAA-positive [119]. Another study followed 32 patients with PAF over 4 years, and found positive seeding in 94% of cases [120]. Five samples with a lower maximum fluorescence and elevated neurofilament levels later developed MSA, while four others developed PD and DLB, suggesting that these prodromal stages might already present α-syn strains that are committed to develop a specific α-synucleinopathies. However, previous autopsy reports of PAF describe the presence of LBs within the autonomic nervous system, which sometimes progress to the substantia nigra and, even after long-term follow-up, only about a third of patients have been shown to phenoconvert to classical α-synucleinopathies [7,121]. It is, therefore, plausible that LB pathology can remain limited to selected brain areas, although the question of how and why requires further study. This is interesting, as it may imply that the relationship between LB pathology and α-syn seeding is not straightforward, and neither is the mechanism. The observable seeding ability might be attributed to α-syn species not readily detected by immunohistochemistry, such as α-syn oligomers.

#### 3.3.3. The Potential Synergistic Effects with Other Prion-like Proteins

Postmortem examinations illustrate the frequent co-occurrence of other pathological prion-like protein deposition with α-syn in the brain. An estimated 40% prevalence of α-syn in combination with tau or Aβ pathology has been documented [122]. This co-occurrence may not be coincidental, as tau and α-syn have been shown to possess a synergistic cross-seeding ability in vitro [123,124]. This ability appears to be influenced by α-syn conformation, which could result from iterations of seeding [125]. If each transmission event of α-syn across connecting cells were like an iteration of seeding, each transmission may also give rise to divergent pathological strains. This hypothesis could shed some light on morphologically different pathological structures in the brainstem and neocortex, or within neuronal and glial cells, across α-synucleinopathies. Moreover, in α-synucleinopathies, the presence of additional tau pathology is associated with a higher cortical α-syn burden, and confers a worse prognosis related to survival and cognitive performance, as well as the interval between the onset of motor and dementia symptoms [122,126]. This may be due to a combination of two strains being able to cross-seed each other, which could result in an accelerated disease course, and increased aggregation. It also suggests that some combinations of strains may not be capable of cross-seeding, and may be responsible for specific cases limited to areas such as the brainstem, or those with a relatively mild disease course.

There is a need to establish the relationship between predominant symptomatology and α-syn strains, as well as which factors may determine a person’s susceptibility to a certain strain and, thus, disease. This would be a useful tool in the implementation of personalized medicine tackling the predisposed symptoms likely to occur. Furthermore, it might readily detect the early stages of disease, even at prodromal stages, thus providing an opportunity to intervene before extensive neurodegeneration has occurred. Understanding the underlying mechanisms will also allow us to understand what triggers and perpetuates α-syn pathology, and how to prevent this.

## 4. Implications of α-Syn ‘Strains’ in Personalized Medicine

Given that mounting evidence indicates that distinct α-syn strains are associated with each of the α-synucleinopathies, those α-syn strains may influence the anatomical vulnerability, disease severity, symptoms, and cellular vulnerability observed. Perhaps, there is not a unique strain in individuals, but rather a coexisting mixture of varying proportions, where the predominant ones could dictate the observable phenotype. Regardless, these strains of α-syn must share similar features and, just like the disease they cause, may not represent unique entities but, rather, a spectrum.

At present, there are no disease-modifying therapies for α-synucleinopathies, but medication or surgery can provide palliative treatment, ameliorating motor symptoms and, to a lesser extent, non-motor symptoms. From a therapeutical point of view, α-syn strain-specific therapies may enable novel translational paradigms, with a personalized medicine approach, in the treatment of α-synucleinopathies that would be more effective than the non-specific targeting of α-syn aggregation. This enables the targeting of the mechanism leading to the formation of pathological α-syn aggregates, in a disease-specific manner. Likewise, the discovery of techniques capable of specifically recognizing individual α-syn strains may help to provide an accurate diagnosis in living patients, and predict the disease course. This could lead to a decreased rate of misdiagnosis, which may help to better stratify patient populations prior to clinical trials, increasing the accuracy of the results and conclusions obtained.

Since diagnosis is only confirmed postmortem, the ability to address the adequate treatment for each patient during their disease course is limited. Current efforts focus on optimizing seeding assays, which hope to allow for a non-invasive form of detecting α-synucleinopathies antemortem. This may allow for a more accurate and early diagnosis, possibly even in the prodromal stage. This development would be revolutionary, as it would identify groups with an increased risk, and provide the option of prophylactic neuroprotective treatment prior to the extensive neurodegeneration associated with parkinsonian symptoms. Non-motor symptoms, such as constipation and RBD, are known to appear years before motor symptoms, so understanding the real cause of these common symptoms may help alter treatment guidelines accordingly. However, it remains to be determined whether PAF and RBD represent additional α-syn strains, or whether they simply constitute early phases of the classical α-synucleinopathies. This raises the question as to when the final misfolding into a disease-causing α-syn strain occurs, and commits an individual to the development of a specific disease phenotype, as well as whether this process is stochastic, or influenced by factors extrinsic to α-syn. Curiously, it may be possible that the introduction of another α-syn strain associated with a better prognosis, or external factors associated with the development of one, could cause evolution into a milder disease course, or the amelioration of symptoms.

## 5. Conclusions

Our understanding of the existence of α-syn strains, as well as misfolding, aggregation, and dissemination throughout the central nervous system, has gradually increased over the years. However, further research is needed to determine the spectrum of α-syn structures present both within and amongst individual α-synucleinopathies. The strain hypothesis reviewed here may contribute to the explanation of the complex clinical picture associated with α-synucleinopathies. The conformation of each strain significantly contributes to their biological, chemical, and physical properties, which we have shown may influence pathology, from the cellular to the disease level. Understanding how different α-syn conformations attack specific brain regions and cells in a disease-specific manner holds incredible potential for insights into understanding the widespread neuronal death and dysfunction. Furthermore, an awareness of which regions are vulnerable to which α-syn strains offers an opportunity to develop ways to protect neuronal populations at risk. This knowledge would not only prove invaluable in understanding the mechanistic underpinnings of α-synucleinopathies, but could also extend to other neurodegenerative prion-like proteins, such as tau, Aβ, and TDP-43.

## Figures and Tables

**Figure 1 ijms-24-13199-f001:**
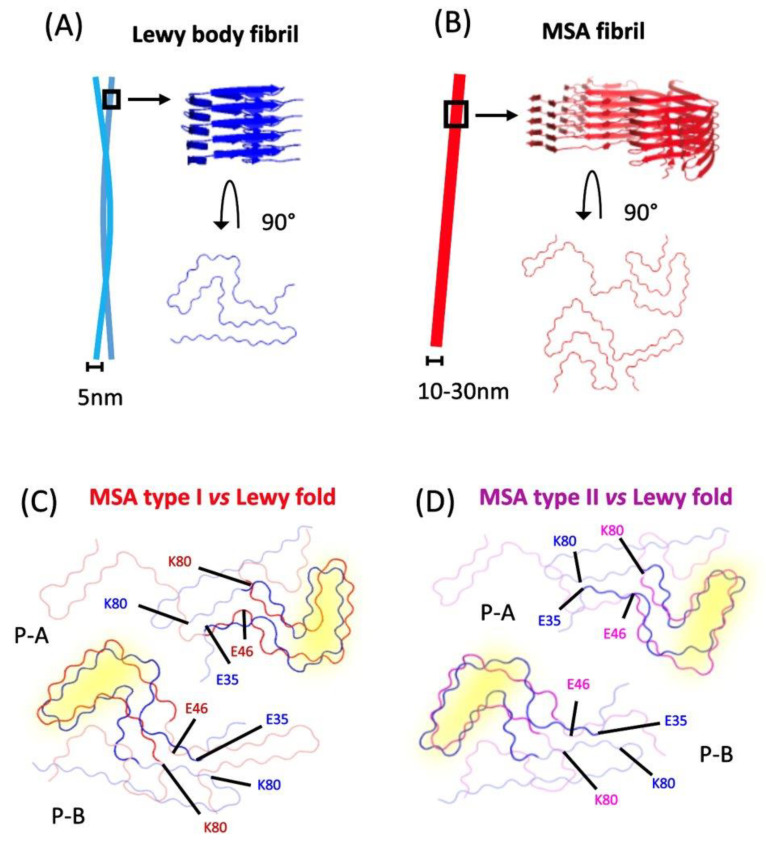
Structural differences between strains. (**A**) Fibrils in LBs are commonly associated in pairs, and stack on top of each other to form a twist, with a diameter of 5 nm. (**B**) Fibrils found in GCIs consist of α-synuclein dimers that can form a straight or twisted structure, with a diameter ranging from 10 to 30 nm (MSA type I dimers are shown as a generic example). (**C**,**D**) α-synuclein monomers found in MSA type I and II (Protein Data Bank (PDB) atomic model accession numbers: 6XYO & 6XYP), respectively, are superimposed to the Lewy body fold (PDB: 8A9L), displaying regions with structural similarities (highlighted in yellow). Key amino acid residues are labelled, showing that different interactions with the residue K80 are associated with either the LB or MSA folds [57,64].

**Figure 2 ijms-24-13199-f002:**
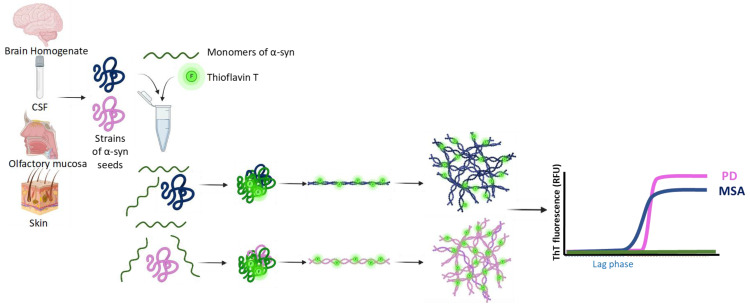
Scheme depicting the methodology for the amplification of α-syn seeds via seeding amplification assays (SAAs). A schematic illustrating the sources of seed-competent α-syn oligomers, which are combined with α-syn monomers and Thioflavin T (ThT) during SAAs. The α-syn seeds recruit the monomers, misfolding them, and encouraging them to aggregate into fibrillar structures. ThT fluorescence is recorded during the assay, resulting in a curve that is initially low in lieu (lag phase), but increases as misfolded α-syn monomers begin to aggregate (exponential phase) and, eventually, reaches a plateau. Evidence suggests that different strains of a-syn are likely to undergo this process, with variations in the seeding kinetics and terminal structures. The different α-syn strains present in MSA and PD display different seeding kinetics, with MSA α-syn aggregating faster, but reaching a lower plateau, in comparison to PD α-syn [29].

## Data Availability

Not applicable.

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
