# Peer review of "Beyond Strains: Molecular Diversity in Alpha-Synuclein at the Center of Disease Heterogeneity"

_ijms, 2023, doi:10.3390/ijms241713199_

Round 1

Reviewer 1 Report

Wojewska et al., have very nicely summarized the molecular mechanisms of α-synuclein heterogeneity amongst related diseases, providing an up-to-date and comprehensive overview of the current understanding of such heterogeneity and its clinical and pathological phenotypes. Moreover, it includes a concise summary of cutting-edge detection assays/technologies used to achieve this understanding. The review is well-structured and logically sound. 

Minor changes suggested:

1. Line 29-30: Need a reference

2. Line 35-38: Long sentence 

3. Line 52-55: Need a reference

4. Line 128-131 Ref{23-26} are not from ‘recent work’

5. Line 155: there seems to be an extra ‘s’ before ‘α-syn species’

6. Line182: double full stops

7. Please check the font of figure legend 1; it doesn't look consistent in the version I received.

8. In Figure 2, there are no 'a' or 'b' labels, but they are present in the legend.

9. Line 597-601: Long sentence

10. I would remove the bullet points of the level 4 subtitles

11. From Lind 189, shall the “Clinical correlation’ be“3.2.3 Clinical correlation”? It seems to me that it can be a stand-alone section.

12. Check the font of the Conclusion paragraph. 

Author Response

Dear Editor,

Thank you for giving us the opportunity to submit a revised draft of our manuscript titled ‘Beyond strains: Molecular diversity of alpha-synuclein at the center of disease heterogeneity’ to the International Journal of Molecular Science. We appreciate the time and effort that you and the reviewers have dedicated to providing valuable feedback on our manuscript. We are grateful to the reviewers for their insightful comments on our paper. We have been able to amend the manuscript to reflect most of the suggestions provided by the reviewers. The amended changes have been highlighted. Please also note that since the line numbers have changed with the modifications, we provide the new line numbers to index our replies below.

Here is a point-by-point response to the reviewers’ comments and concerns. All comments have been actioned unless stated otherwise.

Reviewer 1:

Minor changes suggested:

  • Line 40 (previous line 29-30): Reference added.
  • Line 44-47 (previous line 35-38): This long sentence was reworded and split into two sentences.
  • Line 60 (previous line 52-55): Reference added.
  • Line 120 (previous line 128-131): Wording changed to clarify that the studies are not recent.
  • Line 133 (previous line 155): The extra ‘s’ before ‘α-syn species’ was removed.
  • Line 154 (previous line 182): The double full stop was changed to a single full stop.
  • The font of Figure legend 1 was formatted.
  • Figure 2 has had the a and b labels removed to yield a single unified figure.
  • Line 494-497 (previous line 597-601): This long sentence was reworded and split into two sentences.
  • The bullet points of the level 4 subtitles have been removed.
  • Added section 3.2.3 Clinical correlation as a stand-alone section.
  • Formatted the font of the Conclusion paragraph.

Reviewer 2 Report

Review of the manuscript by Wojewska et. al. Beyond strains: Molecular diversity of alpha-synuclein at the center of disease heterogeneity.

This review manuscript is, in my opinion, overdone and rather chaotic. Several other review articles have been published in recent years on similar topics (below), which seem easier to follow and provide clearer picture of the problem. The authors additionally include in their manuscript the prion hypothesis in the development of synucleinopathies, but this does not improve the substantive quality of the manuscript, but rather makes the article messy and overcomplicated.

https://www.ncbi.nlm.nih.gov/pmc/articles/PMC8301881/

https://pubmed.ncbi.nlm.nih.gov/32440702/

https://link.springer.com/article/10.1007/s00401-020-02163-5

https://www.frontiersin.org/articles/10.3389/fnagi.2022.907293/full

https://onlinelibrary.wiley.com/doi/full/10.1111/jnc.14965

https://www.frontiersin.org/articles/10.3389/fneur.2021.737195/full

https://pubmed.ncbi.nlm.nih.gov/29704213/

https://pubmed.ncbi.nlm.nih.gov/28751258/

For more detailed comments and/or questions, please see the pdf file of the original manuscript (attached).

Author Response

Dear Editor,

Thank you for giving us the opportunity to submit a revised draft of our manuscript titled ‘Beyond strains: Molecular diversity of alpha-synuclein at the center of disease heterogeneity’ to the International Journal of Molecular Science. We appreciate the time and effort that you and the reviewers have dedicated to providing valuable feedback on our manuscript. We are grateful to the reviewers for their insightful comments on our paper. We have been able to amend the manuscript to reflect most of the suggestions provided by the reviewers. The amended changes have been highlighted. Please also note that since the line numbers have changed with the modifications, we provide the new line numbers to index our replies below.

Here is a point-by-point response to the reviewers’ comments and concerns. All comments have been actioned unless stated otherwise.

Reviewer 2:

Abstract

  • Line 19: DLB and MSA were changed to lowercase letters. This change was also implemented in line 41.
  • Line 26: a-Syn was changed to Strains of a-syn to avoid capitalizing Syn.
  • Line 27-28: This sentence was rewritten to improve flow and clarity.

Introduction 1.1

  • Line 42: Source of this information was added here.
  • Line 47: Newer reference cited here.
  • Line 51: The double space was removed.
  • Line 62: The double space was removed.
  • Line 64: Source of this information was added here.
  • Line 66: Source of this information was added here.

Section 1.2

  • This chapter has been rewritten and restructured. We wanted to introduce the prion hypothesis as we believe it is fundamental in understanding the concept of strains. A lot of the experimental evidence we explore throughout section 2 assumes the prion-like nature of a-syn and was originally used to establish the existence of prion strains. However, we understand that it remains controversial how to what degree a-syn strains as well as strains of the other neurodegenerative proteins may resemble prion strains.
  • Line 105: Missing space was inserted.
  • Line 105: Sentence was reworded.
  • Line 118-119: referring to ‘Akin to the prion protein, α-syn is suspected to form different strains with different properties which may underlie differences seen across α-synucleinopathies’ The reviewer says ‘Why suspected? The paper by Shahnawaz (citation No. 29) states that: "Aggregates of α-synuclein in distinct synucleinopathies have been proposed to represent different conformational strains of α-synuclein that can self-propagate and spread from cell to cell" Therefore, we believe we could do that statement.
  • Line 120: Wording changed to clarify that the studies are not recent.
  • Line 127: This section that was previously line 140-147 was removed to improve the flow of chapter.

Section 2.1

  • Line 136: Source of this information was added here.
  • Line 141: Missing space was inserted.

Section 2.2

  • Line 168: Missing space was inserted.
  • Line 177: Changed the wording of this sentence as it was meant to introduce the content of the next 5 sub chapters.

Section 2.2.1

  • Line 184-185: The reviewer questioned “Not only these two enzymes. Isn't it general feature of enzymes????” regarding our statement that “Proteinase K and thermolysin digest proteins based on the exposure of cleavage recognition sites within the 3D structure of the protein”. This comment was not actioned. Not all enzymes cleave proteins. Although cleaving peptide bonds is a feature common to all proteases, we found it important to mention the specificity of these two enzymes that we focus on. They are not promiscuous enzymes unlike, for example trypsin, and this explains why these enzymes yield distinct fragments upon proteolysis of a-syn derived from different sources.
  • Line 187: Source of this information was added here.
  • Line 189: Source of this information was added here.
  • Line 190: Source of this information was added here.

Section 2.2.2

  • Line 229: We have amended our wording to clarify that the paper concerns the prion protein, not alpha-syn. It was not our intention to mislead the reader but illustrate that significance of the original studies on prion strains and how they influence studies on a-syn strains currently.
  • Line 232: This sentence was intended to introduce the main papers of the section. It has been amended with the main papers of the section cited.
  • Line 262: Sentence was reworded to correct grammar.
  • Line 270: The reviewer suggested changing from ‘a’ to ‘an’ in the sentence “Interestingly, the structural commonalities of PD and DLB α-syn filaments suggest that a unifying α-syn strain may underlie them”. This comment has not been actioned as we do not agree with the grammatical change proposed as since the word unifying starts with a consonant sound (you), therefore it takes the indefinite article “a”. The rule for using ‘a’ or ‘an’ is based on pronunciation, not spelling.
  • Line 275: Figure formatting fixed.
  • Line 275: The legend of figure 1 was amended to clarify that it illustrates the a-syn fibrils found in both LBs and GCIs.

Section 2.2.3

  • Line 289: This comment has not been actioned as the reference is cited at the end of the sentence.
  • Line 290: More references added.
  • Line 296: Figure 2 has had the a and b labels removed to yield a single unified figure.
  • Line 298-304: Experimental details removed to provide a more general view of the RT-QuIC assay since there is not a consensus protocol yet.
  • Line 316: The double space was removed.
  • Line 345: The double space was removed.
  • Line 353: The double space was removed.

Section 2.2.4

  • Line 363: Reference was added.
  • Line 385: Source of information was added in the following line.
  • Line 390: Missing space was added.
  • Line 391: The source of this information was cited again to improve clarity.

Section 2.2.5

  • Line 414: The double space was removed.
  • Line 415: Missing space was inserted.
  • Line 416: The comment was not actioned; the reference was not added as the reference is explained (and referenced) in the following sentence.
  • Line 425: The double space was removed.
  • Line 493: The double space was removed.

Section 3

  • Line 499: The blue title was formatted.
  • Line 510: The double space was removed.
  • Line 553: The double space was removed.
  • Line 560: The blue title was formatted.
  • Line 577: The double space was removed.
  • Line 610: The double space was removed.
  • Line 620: The double space was removed.
  • Line 631: The double space was removed.

Conclusion

  • Line 729-740: Fonts have been formatted
  • Line 772: Methodology contribution removed.
  • Line 773: The contribution of Jose Guijarro-Nunes was added as Figures.
  • Line 780-783: Information regarding Institutional Review Board Statement, Informed Consent Statement, Data Availability Statement, and Conflicts of Interest added.

Best wishes,

JAA, MJW, MOJ, JGN

Round 2

Reviewer 2 Report

The manuscript was edited, typo errors have been amended and references added where suggested. The explanations given by the authors to my concerns clarified to some extend their intensions behind this review paper.